# Covid19Vaxplorer: A free, online, user-friendly COVID-19 vaccine allocation comparison tool

**Imelda Trejo** [1], **Pei-Yao Hung** [2], **Laura Matrajt** [1,3]*

**1** Vaccine and Infectious Disease Division, Fred Hutchinson Cancer Center, Seattle, Washington, United States of America, **2** Institute For Social Research, University of Michigan, Ann Arbor, Michigan, United States of America, **3** Department of Applied Mathematics, University of Washington, Seattle, Washington, United States of America

☯ These authors contributed equally to this work.
* laurama@fredhutch.org

## Abstract

There are many COVID-19 vaccines currently available, however, Low- and middle-income countries (LMIC) still have large proportions of their populations unvaccinated. Decision-makers must decide how to effectively allocate available vaccines (e.g. boosters or primary series vaccination, which age groups to target) but LMIC often lack the resources to undergo quantitative analyses of vaccine allocation, resulting in ad-hoc policies. We developed **Covid19Vaxplorer** (https://covid19vaxplorer.fredhutch.org/), a free, user-friendly online tool that simulates region-specific COVID-19 epidemics in conjunction with vaccination with the purpose of providing public health officials worldwide with a tool for vaccine allocation planning and comparison. We developed an age-structured mathematical model of SARS-CoV-2 transmission and COVID-19 vaccination. The model considers vaccination with up to three different vaccine products, primary series and boosters. We simulated partial immunity derived from waning of natural infection and vaccination. The model is embedded in an online tool, **Covid19Vaxplorer** that was optimized for its ease of use. By prompting users to fill information through several windows to input local parameters (e.g. cumulative and current prevalence), epidemiological parameters (e.g basic reproduction number, current social distancing interventions), vaccine parameters (e.g. vaccine efficacy, duration of immunity) and vaccine allocation (both by age groups and by vaccination status). **Covid19Vaxplorer** connects the user to the mathematical model and simulates, in real time, region-specific epidemics. The tool then produces key outcomes including expected numbers of deaths, hospitalizations and cases, with the possibility of simulating several scenarios of vaccine allocation at once for a side-by-side comparison. We provide two usage examples of **Covid19Vaxplorer** for vaccine allocation in Haiti and Afghanistan, which had as of Spring 2023, 2% and 33% of their populations vaccinated, and show that for these particular examples, using available vaccine as primary series vaccinations prevents more deaths than using them as boosters.

## Introduction

The COVID-19 pandemic has killed more than 6.8 million people worldwide as of March 2023 [1]. There are more than 20 vaccines currently available with various levels of vaccine

**Data Availability Statement:** All the Data used in this work is publicly available and given in the references. The code is available at: https://github.com/FredHutch/covid19vaccinesimulator.

**Funding:** This work was supported by the National Science Foundation under Grant No.2210382. This work was partially supported by grants from the National Institutes of Health (UM1AI068635). L.M. was also supported by a grant from Centers for Disease Control and Prevention (NU38OT000297-02) through their cooperative agreement with the Council of State and Territorial Epidemiologists. The funders The funders had no role in study design, analysis, decision to publish, or preparation of the manuscript.

**Competing interests:** The authors have declared that no competing interests exist.

effectiveness [2]. High-income countries (HIC) have vaccinated and boosted several times the majority of their eligible populations, but only 28% of the population of low income countries have received at least one dose of vaccine. As of March 2023, some low-income countries (LIC) (e.g. Haiti, Senegal) have less than 10% of their population fully vaccinated [3]. COVAX, a global effort co-led by the Coalition for Epidemic Preparedness Innovations (CEPI), the Global Vaccine Alliance (Gavi) and the World Health Organization (WHO), was established to distribute donated vaccines to low- and middle-income (LMIC) countries. Because LMIC are highly dependent on COVAX, many of these countries have received, and are likely to continue to receive, a portfolio of several vaccine products, with very different efficacies against specific variants, forcing public health officials in these regions to take decisions about how to best allocate these resources. However, many of these countries lack the resources to develop quantitative models to evaluate possible vaccine allocations, resulting in an ad-hoc allocation of precious resources. Mathematical models were extensively used during the pandemic to help public health officials with several aspects of the pandemic response: early on for planning health-care capacity and enacting social distancing interventions (e.g. [4–15], for planning vaccine allocation (e.g. [16–33]), for investigating the effect of lifting social distincincing interventions (e.g. [34–36]), for now- and forecasting (e.g. [37–40]), for evaluating the use of antivirals (e.g. [41–45]), for investigating the effect of new variants and waning immunity (e.g. [46–53]. In addition, some online tools were developed for simulating SARS-CoV-2 transmission [54–56]. However, to our knowledge, these tools are usually centered in a single country, state or region. Moreover, most of the research has been centered around HIC, with fewer examples considering LMIC. Existing online tools either have fixed worldwide epidemic projections (number of deaths, hospital resource use, infections) that are not interactive or they have very limited capabilities for user-input, notably not allowing the user to select multiple vaccines with different efficacies. More importantly, as the pandemic has entered an endemic phase in HIC, and the urgency for modeling vaccine allocation has dissipated in these countries, most of these tools are no longer maintained.

We developed **Covid19Vaxplorer** (https://covid19vaxplorer.fredhutch.org/), a free, user-friendly tool that simulates region-specific COVID-19 epidemics in conjunction with vaccination campaigns with multiple vaccine products with the goal of providing decision makers, especially in LMIC, with an easy-to-use tool for planning vaccine allocation and distribution. The aim of our tool is to provide public health officials in different regions in the world a general framework for vaccine allocation guidance and comparison, allowing users in 183 regions (representing ≥97.2% of the global population) to simulate regional epidemics based on local demographics, epidemiology, logistical constraints, and vaccine availability. The tool produces key outcomes including expected numbers of deaths, hospitalizations and cases, with the possibility of simulating several scenarios of vaccine allocation at once for a side-by-side comparison. The tool is highly flexible and customizable, while remaining very user-friendly.

## Materials and methods

### Model

**Basic model structure.** Based on Matrajt *et al.* (2020) and Matrajt *et al.* (2021), [25, 26], we developed an age-structured deterministic mathematical model of SARS-CoV-2 transmission and vaccination with up to three vaccine products (referred as "vaccines" for simplicity below). The model accounts for infections, reinfections, naturally-acquired and vaccine-induced immunity, primary vaccination series and boosters (single boosters) and waning. Based on the groupings made in several countries ([57] or [58] for example), we divide the population into five age groups: 0–19, 20–49, 50–64, 65–74, and those 75 and older. This

grouping reflects our current knowledge of COVID-19 disease severity, with kids under 18 years old being less affected, and adults being progressively more affected after age 50 [15].

Fig 1 shows the diagram of the model where individuals in each row have increased level of protection. Within each age-group (indexed by $\alpha = i$, $i = 1, 2, \ldots, 5$, but omitted in the diagram for simplicity), individuals can belong to one of the following immune classes:

- Unvaccinated individuals who have never been vaccinated, (Fig 1a). We use the index $\alpha = i$ to refer them, i.e, $S_i$, $E_i$, $A_i$, $P_i$, $I_i$, $H_i$, $RA_i$, $R_i$, $RH_i$, $RRA_i$, $RR_i$, and $RRH_i$.

- Unvaccinated individuals who have been infected and have had their protection waned (indexed by $\alpha = P_i$, Fig 1b). These individuals have partial naturally-acquired protection.

- Individuals who have been vaccinated with a primary series with vaccine product $j$ and have had their vaccine protection waned (indexed by $\alpha = W_i$, Fig 1c). These individuals have partial hybrid protection gained from vaccines and infections.

- Individuals who have been vaccinated with a primary series with vaccine product $j$ (indexed by $\alpha = V_{ij}$, Fig 1d).

- Individuals who have been boosted with vaccine product $j$ (indexed by $\alpha = B_{ij}$, Fig 1e). These individuals have protection from having been vaccinated twice (primary series and booster), or have a combined protection gained from being vaccinated with a primary series and having at least one infection after that.

The model allows for vaccination with up to 3 vaccine products so that $j = 1, 2, 3$.

For each of these immune classes, we simulate the standard SEIR transitions between different disease/infection states. Susceptible individuals ($S_\alpha$) become infected and transition to the exposed compartment ($E_\alpha$), where they are infected but not yet infectious. After a latent period, exposed individuals become infectious, and will either become asymptomatic infected ($A_\alpha$) or pre-symptomatic infected ($P_\alpha$). Asymptomatic infected individuals (who are still infectious to others despite not having symptoms) recover and move to the removed compartment $RA_\alpha$. The model assumes that an age-dependent proportion of the infections will be asymptomatic (S1 Table provide the default parameters, but these can be changed by the user). The model assumes that Pre-symptomatic infected individuals become symptomatic infected ($I_\alpha$). After a few days, symptomatic infected individuals either recover and transition to the removed compartment ($R_\alpha$) or become hospitalized ($H_\alpha$). Hospitalized individuals recover (transition to the removed class $RH_\alpha$), or die and are removed from the population (arrows were omitted from the Fig 1 for clarity). While the overall transitions are identical in each immune state, the rates at which people transition from one compartment to the others are affected by the immune state. For example, people in the partially infected symptomatic compartments ($I_P$) have a lower hospitalization rate than the unvaccinated symptomatic infected ($I$). Fig 1 shows the transition rates for each immune class. This gives rise to a system with 540 homogeneous compartments.

Hospitalization and mortality rates are age-stratified and are based on those given in Ferguson *et al* (2020), [6]. The reproduction number $R_0$, an input given by the user, is used to estimate the transmission rate, $\beta$, which in turn is used to compute the force of infection $\lambda m$. Model equations, full details and default parameters can be found in the Supporting Information (S1 Text, S1, S2 Tables).

*Loss of immunity.* The removed compartments are split into two phases (the second phase is denoted by $RR$) to allow for a slower representation of the loss of immunity: removed asymptomatic ($RA_\alpha$ or $RRA_\alpha$), removed symptomatic ($R_\alpha$ or $RR_\alpha$) and removed hospitalized

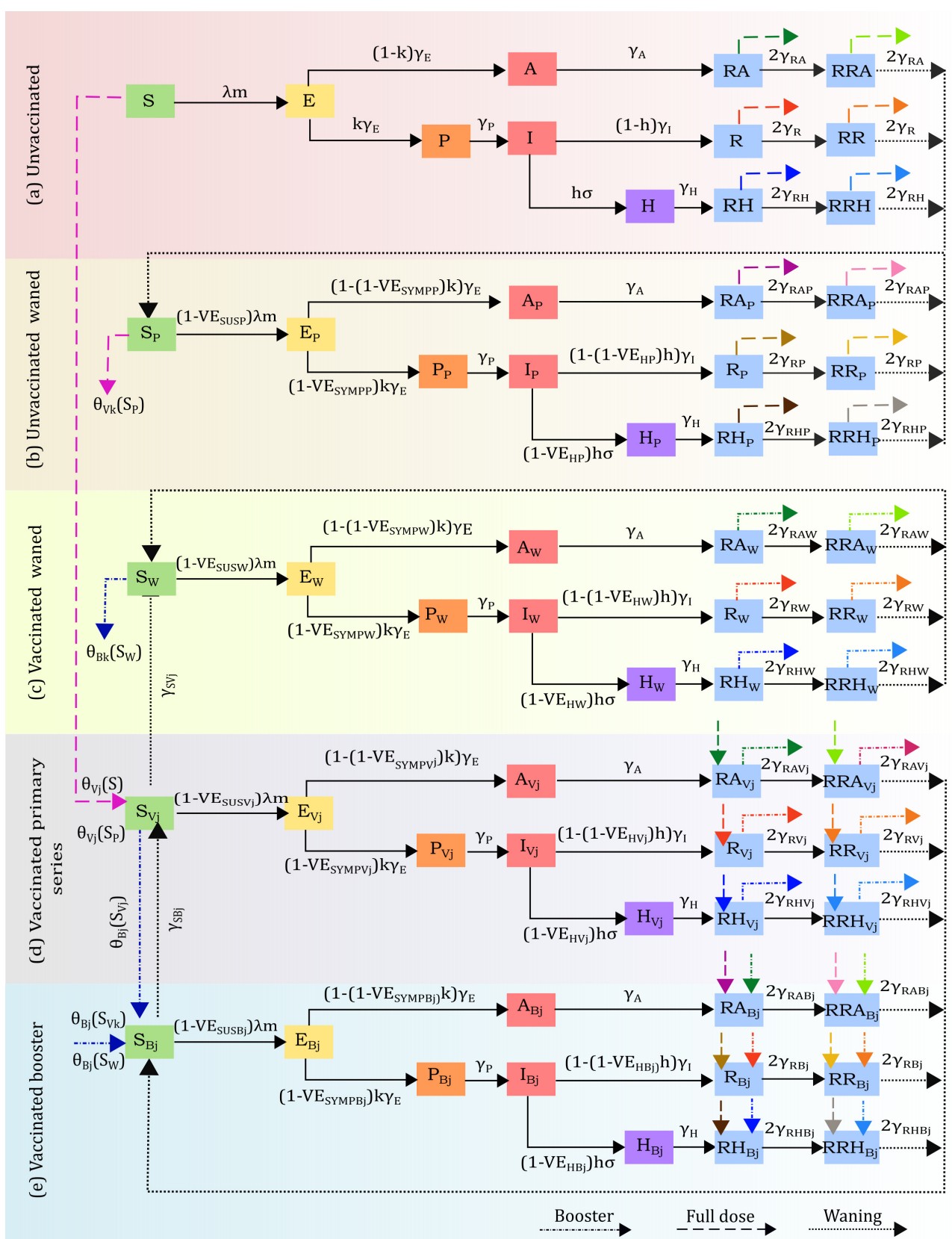

**Fig 1. Flow diagram of the disease progression and vaccine allocation.** Transitions between infection or disease status are depicted with black solid arrows while black dotted arrows describe waning immunity. Dashed and dotted-dashed arrows represent movement between different levels of protection (unvaccinated to vaccinated with primary series to boosted). The colored arrows represent the change in immune state with vaccination. For example, people in the unvaccinated $RA$ compartment move to the corresponding $RA_{V_j}$ compartment in the primary series class. This is depicted by a dark green dashed arrow coming out of $RA$ and going into $RA_{V_j}$. Similarly, compartments that move to the booster compartments are represented by dotted-dashed colored arrows. For example, people in the $R_W$ compartment move to the corresponding boosted compartment, $R_{B_j}$ (represented by a red dotted-dashed arrow coming out of $R_W$ and coming into $R_{B_j}$. The $i$-age group indices are omitted for clarity.

($RH_\alpha$ or $RRH_\alpha$) for $\alpha = i$, $P_i$, $W_i$, $V_{ij}$ or $B_{ij}$. Upon losing their immunity, all individuals in the unvaccinated removed second phase ($RRA_i$, $RR_i$ and $RRH_i$ at rates $2\gamma_{RA}$, $2\gamma_R$ and $2\gamma_{RH}$ respectively) transition to the susceptible partially waned class, $S_{P_i}$. Analogously, individuals in the partially removed second phase classes ($RRA_{P_i}$, $RR_{P_i}$ and $RRA_{P_i}$) also transition (at rates $2\gamma_{RAP}$, $2\gamma_{RP}$ and $2\gamma_{RHP}$ respectively) to the susceptible partially waned class, $S_{P_i}$ as their infection-acquired immunity wanes (Fig 1a and 1b, dotted arrows). Similarly, individuals in the removed classes within the vaccinated waned compartments ($RRA_{W_i}$, $RR_{W_i}$ and $RRH_{W_i}$) transition (at rates $2\gamma_{RAW}$, $2\gamma_{RW}$ and $2\gamma_{RHW}$ respectively) back to the susceptible waned class ($S_{W_i}$) as they lose protection (Fig 1(c), dotted arrows). Individuals who were vaccinated with either a primary series or a booster and got infected move from their respective removed classes ($RRA_{V_{ij}}$, $RR_{V_{ij}}$ and $RRH_{V_{ij}}$ and $RRA_{B_{ij}}$, $RRB_{ij}$ and $RRH_{B_{ij}}$) to the susceptible boosted class, $S_{B_{ij}}$ (Fig 1d and 1e, dotted arrows). Finally, susceptible primary series individuals ($S_{V_{ij}}$) wane their immunity and move to $S_{W_i}$ class (at a rate $\gamma_{S_{V_j}}$) and susceptible boosted individuals ($S_{B_{ij}}$) wane their immunity and move to $S_{V_{ij}}$ class (at a rate $\gamma_{SB_j}$, Fig 1d and 1e, dotted arrows).

*Modeling contacts and social distancing.* In our model, individuals contact each other in four possible locations: home, work, school and community. These locations are represented by four different matrices describing the number of contacts each age group has with other age groups in each of these locations. The number of contacts is then used in the force of infection as the sum of contacts in all locations (see S1 Text for further details). We use the values given in Prem *et al.* (2021) [59] to simulate these contacts. To model social distancing, each of these contact matrices is multiplied by a multiplier $d_l$ ($l = 1, 2, \ldots, 4$), provided by the user, with $0 \leq d_l \leq 1$ (0 represents no contact in this location and 1 represents full contact in this location). The total number of contacts per day matrix $C$ is then obtained as a linear combination of these matrices, where the coefficients are given by the multipliers. For example, if the multipliers for "school" and "work" are set to 0, then this would imply that the total number of contacts does not include any contacts in these locations. The matrix $C$ is used to compute the force of infection in the model (as indicated in equation 6, see S1 Text for complete description).

*Modeling initial conditions.* The user provides the proportion of individuals in each age group who has previously been infected and who is currently infected (previous and current infection prevalence). Because estimating previous infection prevalence might be difficult, **covid19vaxplorer** provides the user with default numbers for the proportion of the population who has been previously infected, and sets this to the one estimated by Barber *et al.* (2022) [60] as of November 2021. Users also provide the proportion of individuals in each age group who has been previously vaccinated with each vaccine product, with either primary series or booster (if none is given, then we assume no one in the population has been previously vaccinated). Taken together, the user needs to provide at least 35 numbers for the initial conditions (1 for the current prevalence, 3 for the previously vaccinated fractions and 3 for the previously boosted fractions in each age group). Because there is potential overlap between these groups

(some people might have been previously infected and vaccinated) we split the population in each compartment in the most parsimonious way as follows:

1. We compute the number of people who are in the intersection of those who have been previously infected and those previously vaccinated in each age group. We then place these individuals in the recovered vaccinated compartments conserving the same proportion of the symptomatic and asymptomatic infections as assumed in the model.

2. We compute the analogous intersection of people who are currently infected and previously vaccinated (with primary series or boosters) in each age group. These people will be placed in the currently infected vaccinated compartments with the following percentages: 24% in the latent compartment, 23% in the pre-symptomatic compartment and the rest in the symptomatic compartment (this was calculated assuming that an infectious individual who will develop symptoms spends on average 2, 1.5 and 5 days in the latent, presymptomatic and symptomatic compartments respectively. We computed analogous percentages for the asymptomatic compartments.

3. The remaining vaccinated population is placed in the susceptible vaccinated and susceptible vaccinated waned.

We repeat this procedure with the boosted vaccinated population.

*Modeling vaccine effectiveness.* We assumed leaky vaccines as described in Halloran *et al. (1997)* [61], having three potential effects: a reduction in the susceptibility to infection $VE_{SUS}$, a reduction in the probability of developing symptoms given infection $VE_{SYMP}$ and a reduction in the probability of developing severe symptoms given symptomatic infection $VE_{SEV}$. In addition, we assume that those with waned immunity (either naturally-acquired or vaccine-induced immunity) would retain some protection against infection, symptomatic infection and/or severe infection.

**Modeling vaccination campaigns.**   *Overall vaccination campaigns.* Our model simulates vaccination campaigns with a primary series or boosters. When considering primary series, if the vaccine product requires two doses, we do not model each individual dose. Vaccinated individuals are immediately protected upon receiving their vaccine. In the following sections we describe in more detail how the vaccination campaigns are modeled.

There are three levels of vaccine distribution:

1. **Vaccine allocation strategy:** The user determines the vaccine allocation strategy: who will get the vaccine, how much vaccine that group will get, and how this vaccine will be used (primary series or booster if possible).

2. **Order of vaccination among selected groups:** Once the allocation is determined, the vaccine is allocated daily among selected groups.

3. **Vaccine distribution among compartments:** Once the allocation strategy is determined, the vaccine needs to be distributed among compartments in a given age/vaccinated group.

*Vaccine allocation strategy.* The user provides the model with the age groups to be vaccinated, the percentage of vaccine allocated to them, and the use of the vaccine, primary series or booster (if possible). If the vaccine is used as a booster, then the user needs to provide for each age group, which previously vaccinated subgroup will get the booster. For example, if 35% of the people in the oldest age group have been vaccinated with Cansino and 40% of the people have been vaccinated with Moderna, then the user needs to decide if the boosters will be applied to those previously vaccinated with Cansino (and what percentage of them will get a booster), with Moderna (and again, the percentage of them to get boosted) or both.

*Order of vaccination among selected groups.* Once that allocation strategy is determined, all age and vaccinated groups receive the vaccine at the same time, proportionally to the proportion of the vaccine allocated to them. For example, assume that we have 1,000,000 doses of vaccine of type *j*, and we allocate 30% of them to group 1 and 70% of them to group 2, all for primary series. If we assume that we can deliver 10,000 doses per day, it will take 100 days to complete our vaccination campaign. In our model, both groups 1 and 2 will get vaccinated each day for 100 days, proportional to the vaccine distribution, that is, group 1 would get 3,000 doses per day, while group 2 would get 7, 000 doses per day (30% and 70% for the daily number of doses respectively).

*Vaccine allocation implementation.* Because this is a deterministic ODE model, time is assumed to be continuous. To implement vaccination campaigns, we proceed as follows: we compute the vaccination rate function (as given in equation 7 in S1 Text) and the start and end of each vaccination campaign. The vaccination rate function will compute for each vaccine, the number of vaccines to be allocated per day from the start of the vaccination campaign till the last day of the vaccination campaign. This vaccination rate function is included in the differential equations (equations 1, 4 and 5, in S1 Text) so that vaccinated individuals (either with a primary series or with boosters) are removed from the other compartments and placed in the corresponding vaccinated compartments (see paragraph below).

*Vaccine distribution among compartments.* We assume that infected symptomatic individuals will not be vaccinated. Hence, once a specific vaccine allocation strategy among the age groups and the previously vaccinated groups has been determined, the vaccine for each group is distributed among all the eligible compartments (susceptible $S_\alpha$, exposed $E_\alpha$, asymptomatic infected $A_\alpha$, pre-symptomatic infected $P_\alpha$, and all the recovered compartments: $RA_\alpha$, $RRA_\alpha$, $R_\alpha$, $RR_\alpha$, $RH_\alpha$, $RRH_\alpha$, where $\alpha = i$, $P_i$, $W_i$, $V_{ij}$ or $B_{ij}$ with $i = 1, 2, \ldots, 5$ and $j = 1, 2, 3$) proportionally to their relative size at that point in time for that age group. This is equivalent to assume that individuals in a population will be randomly vaccinated, except for those who have a symptomatic current infection, as these individuals are typically not allowed to be vaccinated. Infected individuals from compartments $E_\alpha$, $A_\alpha$, and $P_\alpha$ are vaccinated, but the vaccine has no effect on them. For example, if the user has allocated 50% of the vaccine *j* as primary series for children and 50% of the vaccine *j* as booster for the oldest age group in those who have previously gotten vaccine *k*, then we would distribute 50% of the vaccine among group 1 in all the unvaccinated classes except the infected symptomatic ones ($S_1$, $E_1$, $A_1$, $P_1$, $RA_1$, $RRA_1$, $R_1$, $RR_1$, $RH_1$, $RRH_1$) and the other 50% of the available vaccine *j* would go to analogous compartments among the oldest age group (group 5) who received the vaccine *k* previously ($S_{V_{5k}}$, $E_{V_{5k}}$, $A_{V_{5k}}$, $P_{V_{5k}}$, $RA_{V_{5k}}$, $RRA_{V_{5k}}$, $R_{V_{5k}}$, $RR_{V_{5k}}$, $RH_{V_{5k}}$, $RRH_{V_{5k}}$).

*Duration of vaccination campaign/simulation.* We calculate the duration of the vaccination campaign according to the daily vaccination rate provided by the user. Because we can model up to three vaccine products, we will obtain up to three durations for the vaccination campaigns. We then determine the duration of the simulation (provided in the tool as a default, but the user can augment it) as the maximum of these three durations. For example, assume we have three vaccine products: 1,000,000 doses of vaccine 1 starting to vaccinate on day 1, 1,500,000 doses of vaccine 2 starting to vaccinate on day 20, and 800,000 doses of vaccine 3 starting to vaccinate on day 70. Assume that the vaccination rate is the same for all three vaccine products, so that we can distribute 10,000 doses of each every day. Then, it will take 100 days to use all doses of vaccine 1, 150 days to distribute all of vaccine 2, and 80 days to distribute all of vaccine 3. This means that the end of our simulation will be given by the maximum of the end of the vaccination campaigns for each vaccine product, so it would be the max(100, 20+ 150, 80+ 70) = max(100, 170, 150) = 170.

*Booster campaigns*. In our model, all individuals who have been previously vaccinated and receive a booster are moved to the corresponding boosted classes. That is, irrespective of their current immune state (waned or not) we assume that a booster will renew their protection to its fullest. For example, individuals in the compartments $S_{V_{ij}}$ and $S_{P_i}$ who are boosted with vaccine product $k$ would go to the booster compartment $B_{ik}$. We do not distinguish between different vaccines for primary doses in boosted individuals and assume that only the booster defines the protection of the individual. That is, if individual A was vaccinated with vaccine $j$ and individual B was vaccinated with vaccine $k$ and both A and B receive a booster with vaccine $l$, then both A and B are equally protected with the protection conferred by vaccine $l$.

*Immunity conferring events*. Individuals who got a single infection and are vaccinated are moved to the corresponding compartments in the Primary series class. For example, when individuals in the recovered classes $RA_i$, $RRA_i$, $R_i$, $RR_i$, $RH_i$ and $RRH_i$, with $i$ = 1, 2, . . ., 5 get vaccinated with vaccine product $j$, they move to the corresponding compartments $RA_{V_{ij}}$, $RRA_{V_{ij}}$, $R_{V_{ij}}$, $RR_{V_{ij}}$, $RH_{V_{ij}}$ and $RRH_{V_{ij}}$. Unvaccinated individuals who have had a single infection for whom immunity has waned (those in the $S_{Pi}$ compartments) are transferred to the corresponding susceptible vaccinated compartment upon vaccination, such that those vaccinated with a primary series of product $j$ will be transferred from $S_{P_i}$ to the $S_{V_{ij}}$ compartment. The transitions across these classes are depicted in Fig 1a and 1d with single colored dashed arrows (example: $RA_i$ has a green single dashed arrow going out and $RA_{V_{ij}}$ has the corresponding arrow coming in). However, we assume that two infection events act as a primary series vaccination [62, 63]. That is, those individuals who have had two infections and get vaccinated with vaccine product $j$ (those in the $RA_{W_i}$, $RRA_{W_i}$, $R_{W_i}$, $RR_{W_i}$, $RH_{W_i}$, $RRH_{W_i}$) compartments are moved to the respective vaccinated boosted compartments ($RA_{B_{ij}}$, $RRA_{B_{ij}}$, $R_{B_{ij}}$, $RR_{B_{ij}}$, $RH_{B_{ij}}$, $RRH_{B_{ij}}$). This is shown in Fig 1 with colored dotted-dashed arrows (e.g. $RA_W$ has a dark green arrow coming out that corresponds with the dark green arrow coming in $RA_{B_j}$.

## Design process

**Covid19Vaxplorer** was developed through a user-centered design (UCD) [64] process with the goal of creating a user-friendly tool for supporting non-technical users, including health officials, to design vaccine allocation strategies that could lead to a better outcome. It was implemented using modern web frameworks, Vue.js [65] and Flask [66]. With the goal of supporting non-technical users such as health officials with this tool, the team, which consists of both human-centered researcher and mathematical modelers who have experience working with health officials, met regularly to gather requirements on the intended utility of this tool and the rationale for having different parameters needed by the model.

Because a large number of parameters are required (191 parameters per strategy), as is usually the case with mathematical models of infectious disease transmission (e.g. [16, 67], we apply the concept of information architecture [68], which emphasizes the importance of organizing information (e.g., parameters in our case) and providing navigation facility in a way that aligns with users' mental model and workflow.

Our team adapted the card sorting technique [68] as a way to build consensus as to how users might conceptually categorize all the parameters needed by the model and to provide a logical flow for specifying different categories of parameters. The team consolidated the

proposed information architecture using paper prototyping [69] and created a low-fidelity prototype (i.e., paper-and-pencil sketches) as an artifact to iterate on the design.

**Covid19Vaxplorer** was designed to avoid saturation and screen tiredness in its users. To achieve this, **Covid19Vaxplorer** helps them navigate different sets of parameters, starting with basic and simple population characteristics, to a more complicated set of parameter definitions that are required to characterize the vaccine availability and prioritization scheme. The proposed interaction flow consists of 4 sections: "Location", "Vaccine", "Vaccine Planning", and "Outcome".

First, the workflow starts with known information about the region of interest, represented as a small set of parameters, including population, the level of social distancing, and current/ past infection statistics.

Second, **Covid19Vaxplorer** prompts users to enter vaccine-related information, which involves a larger number of parameters compared to the first phase, such as the types of vaccines that have been used to date and that will be available for future use, their effectiveness, and the number of people in each age group who were previously vaccinated.

Third, users are invited to specify planning-related parameters, including vaccine availability (what and when), and how vaccines will be allocated to different age groups within the population. Compared to the second "vaccine" phase, here users are expected to actively operationalize their vaccine allocation approaches into the parameters provided by this tool, before seeing the results.

Finally, users are presented with projections of key epidemic outcomes based on the information they input. These outcomes are peak hospitalizations, cumulative number of deaths and epidemic curves (stratified by age group) representing the number of infections, hospitalizations and deaths. They can explore side-by-side variations of vaccine allocation strategies to compare the outcomes and evaluate which strategy is more effective. Users are able to go back to any phase to update parameters so that they can understand how those changes affect the projections associated with each allocation strategy.

In the next section we describe in more detail each of the phases.

## Tool usage

**Covid19Vaxplorer** allows users to input region-specific parameters (types of vaccines, number of doses, vaccination rate, circulating variants, demographics, non-pharmaceutical interventions) and to compare several scenarios of vaccine allocation. Below we describe the tool functionality in full detail (Fig 2).

The tool has 11 main windows divided in four main sections, where each window asks the user to provide specific information:

1. **Location:** In this section, the user provides basic parameters for the region of interest. The section is divided in three windows:

   a. **Region:** User selects the region they want to simulate among 183 regions, and the basic reproduction number $R_0$ (defined as the average number of secondary infections an infected individual will generate in a fully susceptible population). Suggested values of $R_0$ are provided for several SARS-CoV-2 variants, based on the numbers given on Moore *et al.* (2022) [30], (Fig 3).

   b. **Social distancing:** User selects how people are practicing social distancing at each location (home, work, school, community) if people are practicing social distancing, ranging from "fully social distanced" to "not at all" social distanced. These inputs are translated

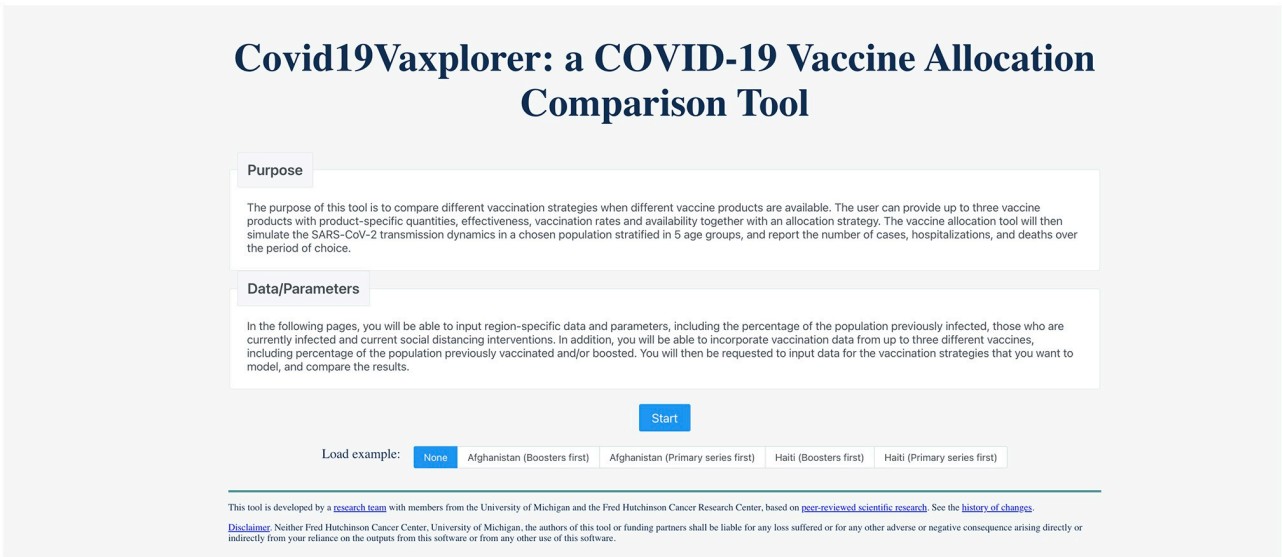

**Fig 2. Landing page for Covid19Vaxplorer.** The user can start a new simulation, import data from their computer or load a preset example.

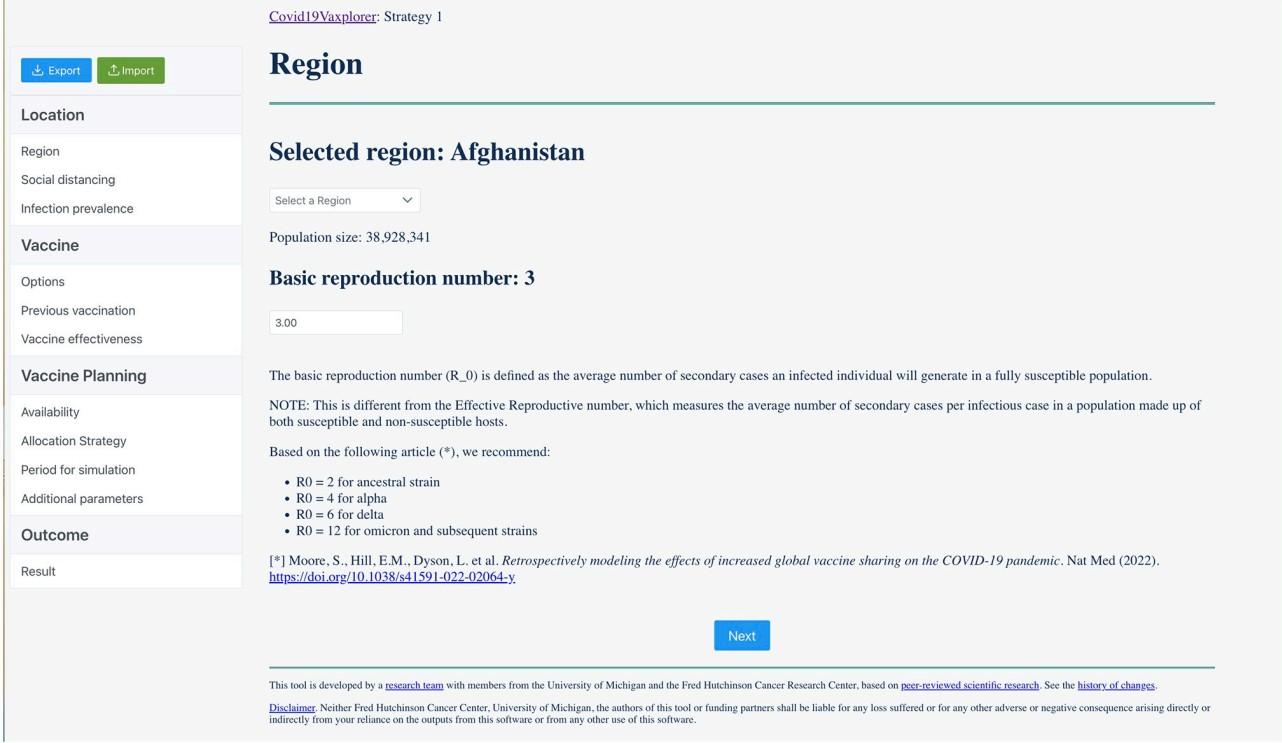

**Fig 3. Example of the region window in Covid19Vaxplorer.** The user can select among 183 regions and they can input a value for the basic reproduction number $R_0$, with some values suggested.

to the model as multipliers (varying from 0 to 1 in 0.1 increments) to each of these contact matrices (see S1 Text for full description, S1 Fig).

 c. **Infection Prevalence:** In this window, the user inputs the proportion of people in each age group who have been previously infected (and who then will have some protection against reinfection, symptoms and/or severe disease) and those who are currently infected. The window provides default percentages for cumulative prior infection as of November 2021 taken from [60] (Fig 4, see section 'Modeling initial conditions" above for full details of how initial conditions are computed).

2. **Vaccine:** The vaccine section of the tool is divided in four separate windows to facilitate the users' input of basic vaccine-related parameters.

 a. **Options:** Here, the user selects up to three vaccine products that will be used in the simulation. The user can choose among 10 pre-loaded vaccine products (the most common vaccine products available in the market for which vaccine effectiveness is readily available) or can create their own vaccine product (Fig 5).

 b. **Previous Vaccination:** In this window, users input for each age group and each vaccine product, the percentage of that group that has been previously vaccinated either with a primary series or a booster. (S2 Fig). If none is given, the model assumes no one in that age group has been previously vaccinated (see section "Modeling initial conditions" above for full details of how initial conditions are computed).

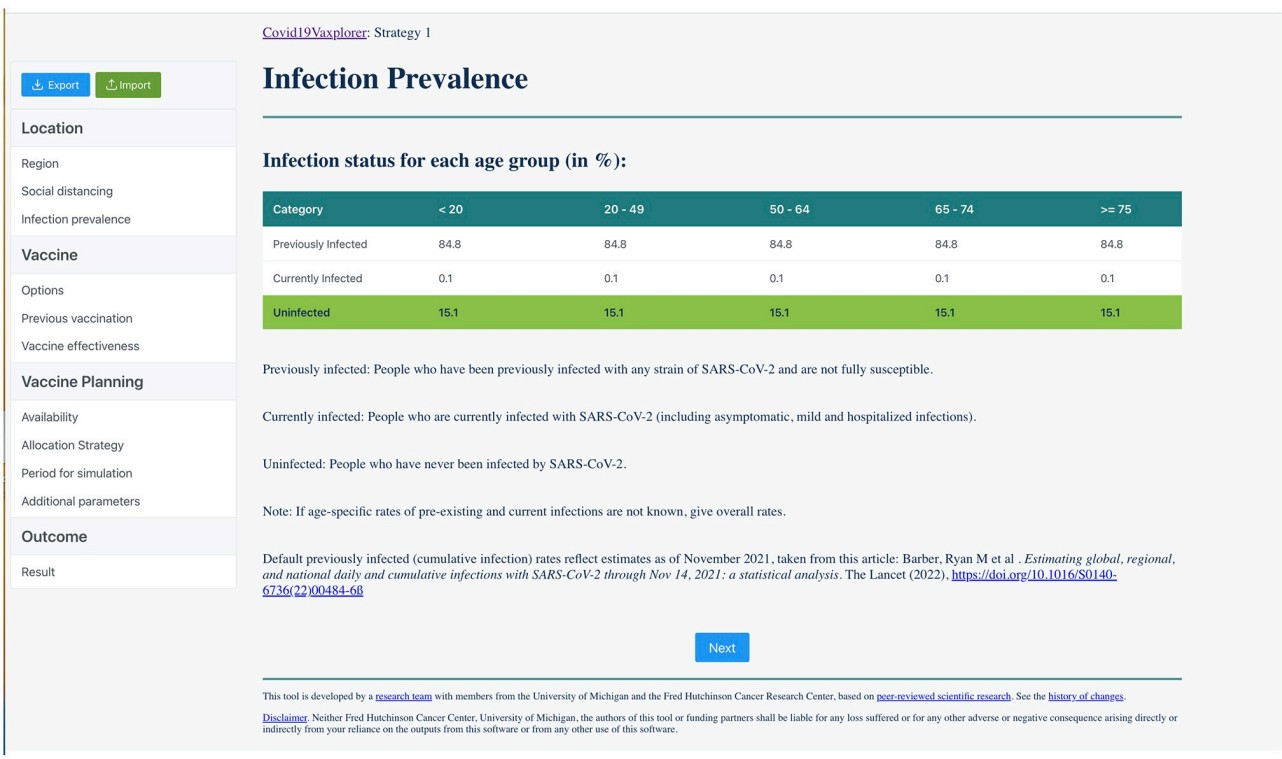

**Fig 4. Example of the infection prevalence window in Covid19Vaxplorer.** The user inputs the percentage of the population in each age group who has been previously infected (cumulative prevalence) and who is currently infected. Default values for the cumulative prevalence are given based on Barber *et al.* (2022) [60].

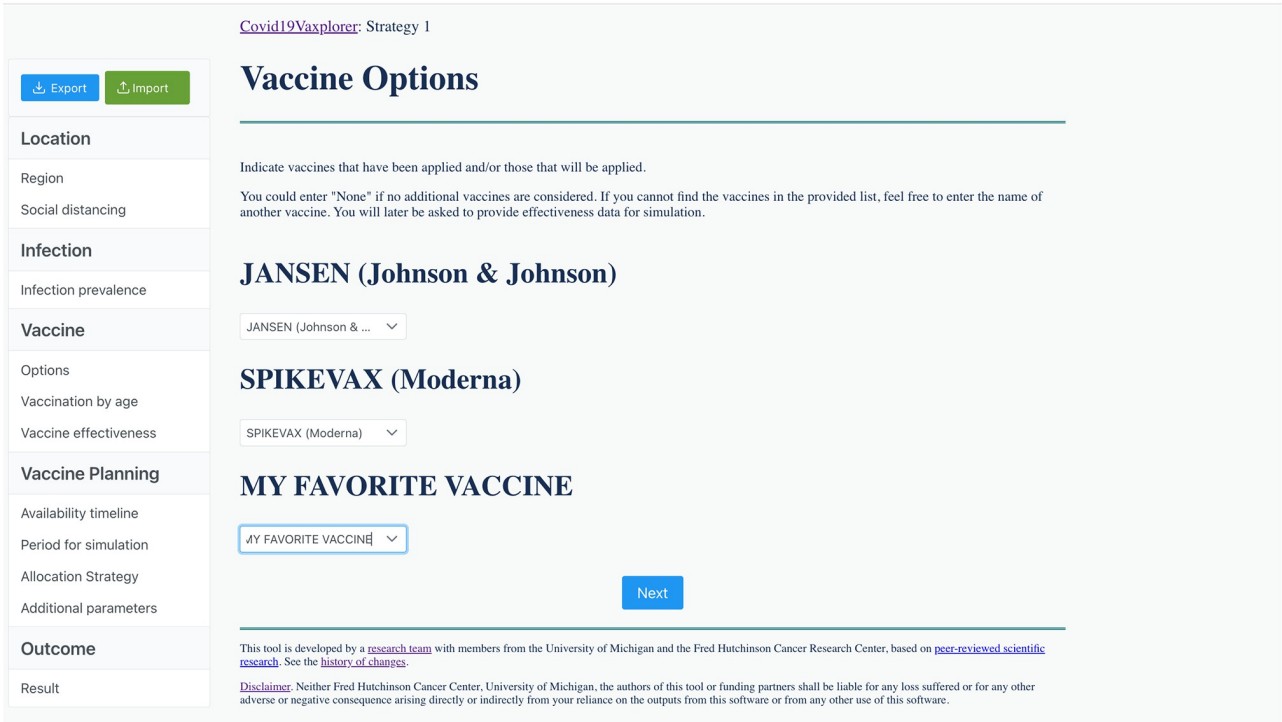

**Fig 5. Example of the vaccine options window in Covid19Vaxplorer.** The user can select up to three vaccine products for their simulation, to choose among 10 pre-loaded vaccines or they can input their own vaccine. For each vaccine product, there is a corresponding vaccine effectiveness value (S2 Table).

c. **Vaccine effectiveness:** Here, the user inputs the vaccine effectiveness for each vaccine product that was previously selected. For convenience, default values are given for the majority of pre-loaded vaccine products, taken from [70] (S2 Table).

3. **Vaccine Planning:** This section is divided in three windows that facilitate the input of data and parameters by the user:

a. **Availability:** The user inputs the amount of up to three vaccine products that will be available (same products used in the "Previously vaccinated" window above) and the number of vaccines that will be administered per day (vaccination rate, S3 Fig).

b. **Allocation strategy:** For each vaccine product, the user is presented with two tables to allocate the available vaccine courses: the primary series table and the booster table. For each table, the user indicates the percentage of vaccine courses to be given as primary series and/or boosters and the date when they will be available. Then, the user further determines the percentage of vaccine to be given to unvaccinated individuals in each age group as primary series (Primary Series table), and the percentage of vaccine to be given in each age group to previously vaccinated individuals as boosters (Boosters table). **Covid19Vaxplorer** includes features like the toolbox button (that resets either the Primary series table or the Booster table for any given vaccine) allowing users to quickly and easily input their vaccine allocation strategy (Fig 6).

c. **Period for simulation:** The user provides the beginning and end dates for the simulation. Default end dates are provided based on the amounts of vaccine available and

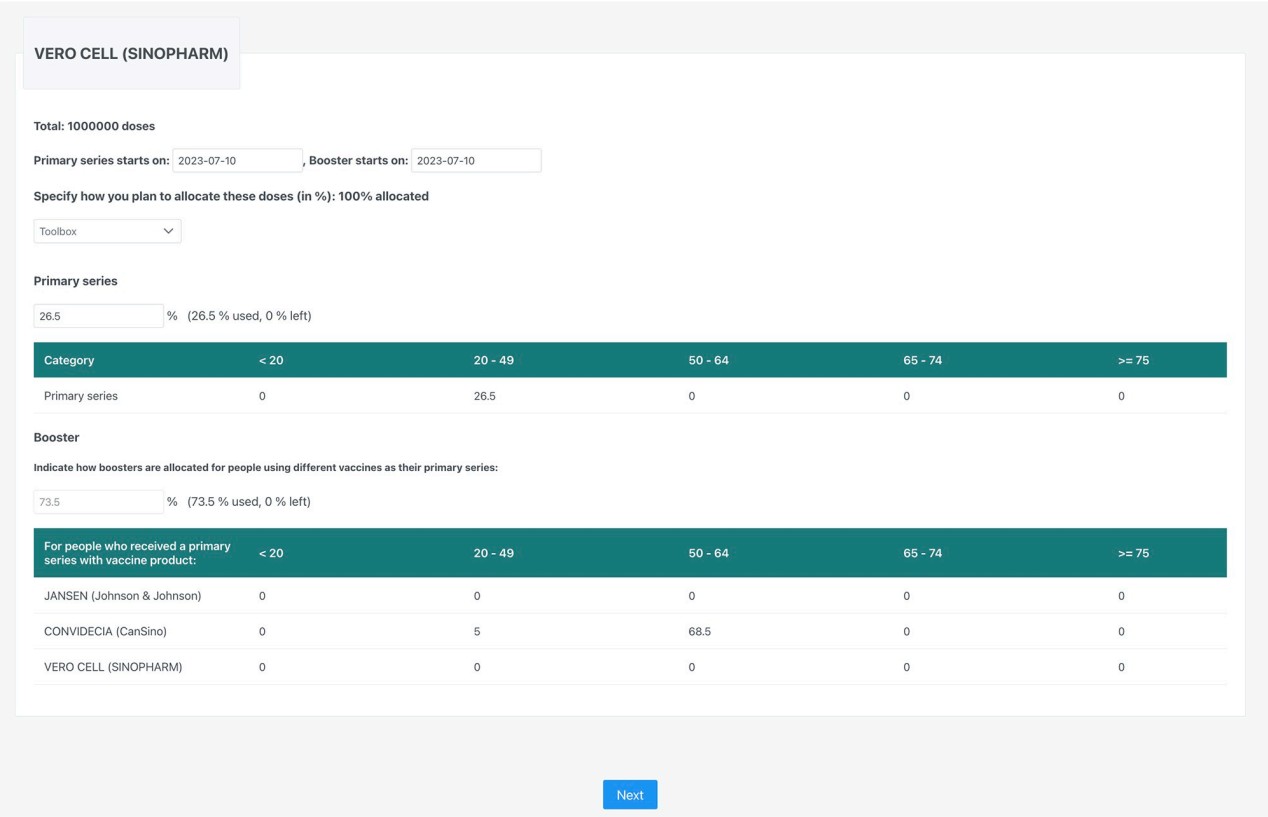

**Fig 6. Example of the vaccine allocation window in Covid19Vaxplorer.** For each vaccine given, the user is presented with two tables to allocate the available vaccine. The user indicates the percentage of available vaccine to be given as primary series and the percentage to be given as boosters. Within each table, the user then determines what percentage of vaccine will be allocated as primary series for each age group, and what percentage of vaccine will be allocated as boosters among age groups and previously vaccinated groups.

vaccination rates, but the simulation end date can be extended beyond that date (full details in section "Duration of vaccination campaign/simulation" above, see S4 Fig).

d. **Additional parameters:** This window presents a table with the additional parameters (e.g. duration of infectious period, duration of naturally-acquired immunity, proportion of asymptomatic infections) needed to run the simulation. Default parameter values are provided (S1 Table and S5 Fig).

4. **Outcome:** This is where the results are displayed and visualized. The following key outcomes are given: Cumulative number of deaths over the simulation period, maximum number of hospitalized individuals, epidemic curves for the daily number of deaths, hospitalizations, symptomatic infections and infections (Fig 7). Each of these outcomes is displayed by age group. These results can be downloaded as a csv file for additonal analysis.

**Covid19Vaxplorer** allows the user to Edit, Duplicate or Delete a given vaccination strategy, and displays the outcomes in plots next to the original strategy, so that comparisons are easily handled. In addition, the user can use the Export button to download the output of the simulation for further analysis.

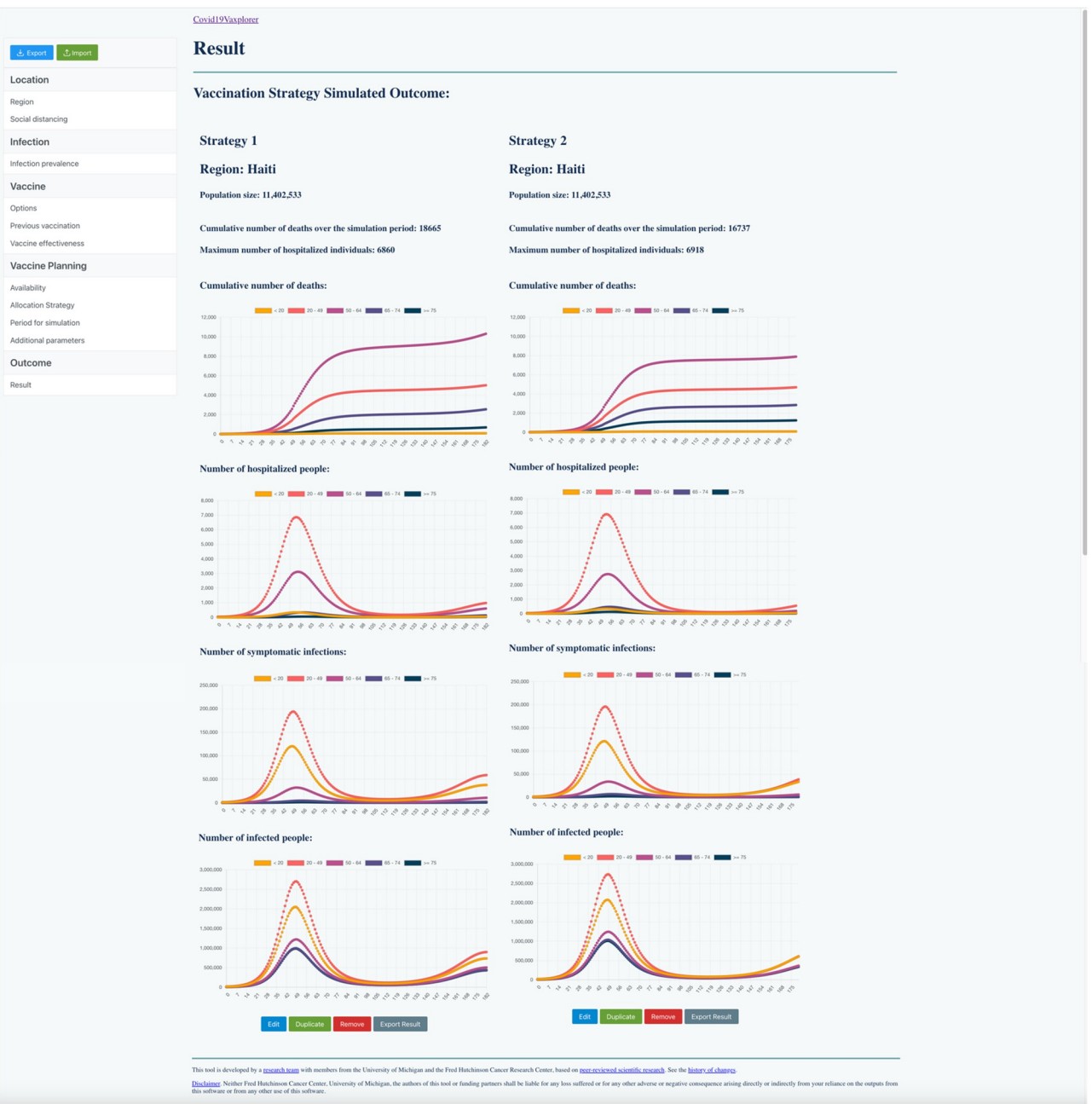

**Fig 7. Covid19Vaxplorer results page for comparing two strategies in a hypothetical situation in Haiti.** Strategy 1, "boosters first" strategy, would result in 18,665 cumulative deaths at the end of the simulation period and 6860 hospitalizations at the peak. In contrast, Strategy 2, "primary series first" would result in 16,737 cumulative deaths and 6,918 hospitalizations at the peak over the same simulation period.

Finally, the user can use the Export/Import buttons to export an entire vaccine allocation strategy for later use or to import an entire vaccination strategy from their computer.

Note that while **Covid19Vaxplorer** provides default parameter values for all the parameters needed to run the model (natural history parameters, vaccine-induced and naturally acquired immunity parameters, cumulative infections, etc), all of these parameters are customizable by the user, so that they can adapt the tool to their particular needs and circumstances (S5 Fig).

## Results

In this section, we provide two examples of the usage of **Covid19Vaxplorer** and how this can help decision-makers. We consider two possible strategies: **Strategy 1**: to give as many boosters as possible (referred below as **"Boosters first"**) and **Strategy 2**: to give as many primary series as possible (referred below as **"Primary series first"**). For the examples below, we consider a 6 month simulation period.

### Hypothetical scenario: Haiti

We explore here a hypothetical scenario for vaccine allocation in Haiti. Based on [60], we assumed that 29.6% of the population in Haiti had been previously infected and have some pre-existing immunity. As of March 2023, 2.06% of the population in Haiti has been fully vaccinated [71]. Since we could not find any specific information about which age groups had been previously vaccinated or about which vaccine products have been previously used, we assumed the following:

- Health-care professionals were prioritized to be vaccinated first. There are approximately 16,000 health care professionals in Haiti [72], and we assumed them to be distributed among the adult age groups proportionally to the population in each of those age groups.

- We assume that the older age groups (75 and older and 65–75 years old) were also prioritized to receive vaccination first as availability permitted after health care workers were vaccinated, starting with the oldest age group.

- Based on news reports, we assume that the first vaccine product given in Haiti was JANSEN.

We consider a hypothetical situation in which Haiti would experience a new SARS-CoV-2 wave with a variant with similar infectivity as the delta variant ($R_0 = 3$) with 0.1% of the population currently infected. We achieve this by inputing in Covid19Vaxplorer a basic reproduction number of 3 in the "Region window" and inputing a current infection of 0.1 in all age groups. It is important to note that this is *not* the effective reproductive number $R_{eff}$, as $R_{eff}$ is computed assuming a population that is not completely susceptible, and for which there might be social distancing interventions in place. The model does not explicitly computes $R_{eff}$, but it is implicitly calculated when the initial conditions and the force of infection are calculated using these factors. We consider that this region would receive an additional 3.5 Million vaccine courses over the following 12 weeks split as follows:

- 1 Million courses of JANSEN vaccine available immediately,

- 1.5 Million courses of CONVIDECIA (Cansino) vaccine available 7 weeks after receiving the SPIKEVAX vaccines,

- 1 Million courses of VERO CELL (Sinopharm) vaccine 4 weeks after receiving the CONVIDECIA vaccine.

We assume that for all three vaccines, 50,000 individuals can be vaccinated daily. For both strategies, we prioritize vaccination of the older age groups. That is, for both strategies, we allocate as much vaccine as possible, either with first doses or boosters to the oldest age group (group 5, 75 years old and older), and if that group is fully vaccinated, we allocate remaining vaccine to group 4 (65–75 years old) and so on. For strategy 1, this results on allocating 20.83%, 1.16%, 0.27% and 1.23% of the Jansen vaccine as boosters to groups 5, 4, 3, 2 respectively, and use the remainder vaccine as primary series, resulting in using 37.02% and 39.49% of the Jansen vaccine as primary series in groups 4 and 3 respectively (no allocation to group 5

since they have been previously vaccinated). Note that this is an arbitrary choice for this current example. It mimics what was observed in reality, where older adults were prioritized to receive the COVID-19 vaccine. S6 Fig shows the percentage of each age group that would be vaccinated with primary series or boosters under each strategy considered.

Using **Covid19Vaxplorer**'s capability to compare strategies back to back, we can determine the best vaccination strategy. We simulated both strategies over a 6 month period. Fig 7 shows the output of **Covid19Vaxplorer** comparing these two strategies. Under the **Boosters first** strategy (Strategy 1) there would be 18,223 cumulative deaths at the end of the simulation period and there would be 6,852 hospitalizations at the peak of the epidemic wave. In contrast, under the **Primary series first** strategy (Strategy 2), there would be 14,553 cumulative deaths at the end of the simulation period and 6,856 hospitalizations at peak. Hence, while the maximum number of hospitalizations is similar, there would be over 3,500 more deaths if a strategy favoring boosters was implemented.

It is likely that the level of pre-existing immunity in the Haitian population is considerably higher than the one considered above. So, we repeated this analysis with all the parameters as above but we considered that 80% of the population in each age group has been previously infected. This analysis showed similar qualitative results with a strategy allocating primary series first outperforming the one prioritizing boosters (S7 Fig). Indeed, under a **boosters first** strategy (strategy 1), there would be over 2,000 more deaths than under a **primary series first** strategy (strategy 2, 8836 vs 6791 deaths) respectively and over 300 more hospitalizations at the peak (2,728 vs 2,451 peak hospitalizations respectively).

## Hypothetical scenario: Afghanistan

Our second example shows a hypothetical scenario for vaccination in Afghanistan. Based on [60], we assumed that 84.8% of the population was previously infected. As of March 2023, 33% of the population had been fully vaccinated in Afghanistan [71]. Based on the UNICEF COVID-19 Market Dashboard [73], 19,724,217 vaccine doses have been assigned to Afghanistan as of February 2023, with a combination of vaccine products: Sinopharm, Vaxzevira, Covaxin, Jansen and an unknown vaccine product [73]. Based on [71], we assumed that 10,754,839 of those have been administered, all of them as primary series. We assumed that vaccine products were administered in the order they were received, and that the oldest age groups were vaccinated first, and younger age groups were vaccinated in decreasing order. S3 Table shows the distribution of previously vaccinated individuals among age groups and vaccine products. For this example, we assume that the difference between the vaccine doses that have been assigned and those that have been administered (6,630,189 doses, all Jansen vaccine) would be administered over the next 6 months, and we compared, as described above, two vaccination strategies: the **"Boosters first" (Strategy 1)** and **"Primary series first" (Strategy 2)** strategies. Because **Covid19Vaxplorer** allows three different vaccine products at a time, we combined vaccine products with similar vaccine effectiveness, so that we formed three groups: Vaccine 1 with Vaxzevira, Vaccine 2 with Sinopharm, Covaxin and Unknown, and Vaccine 3 with Jansen. S8 Fig shows the two hypothetical distributions by age and previously vaccinated groups.

Fig 8 shows a screenshot from the output page of **Covid19Vaxplorer**. In this example, vaccinating as many people with a primary series is still better than using the vaccine in boosters, with 823 less deaths in the former than the latter (23,084 vs 23,907 respectively). Moreover, the **"Primary series first"** strategy would result in a much lower hospitalization peak: 5,647 hospitalizations at peak compared to 8,519 for the **"Boosters first"** strategy. This could be important because it might allow public health officials to maintain the healthcare system within working capacity.

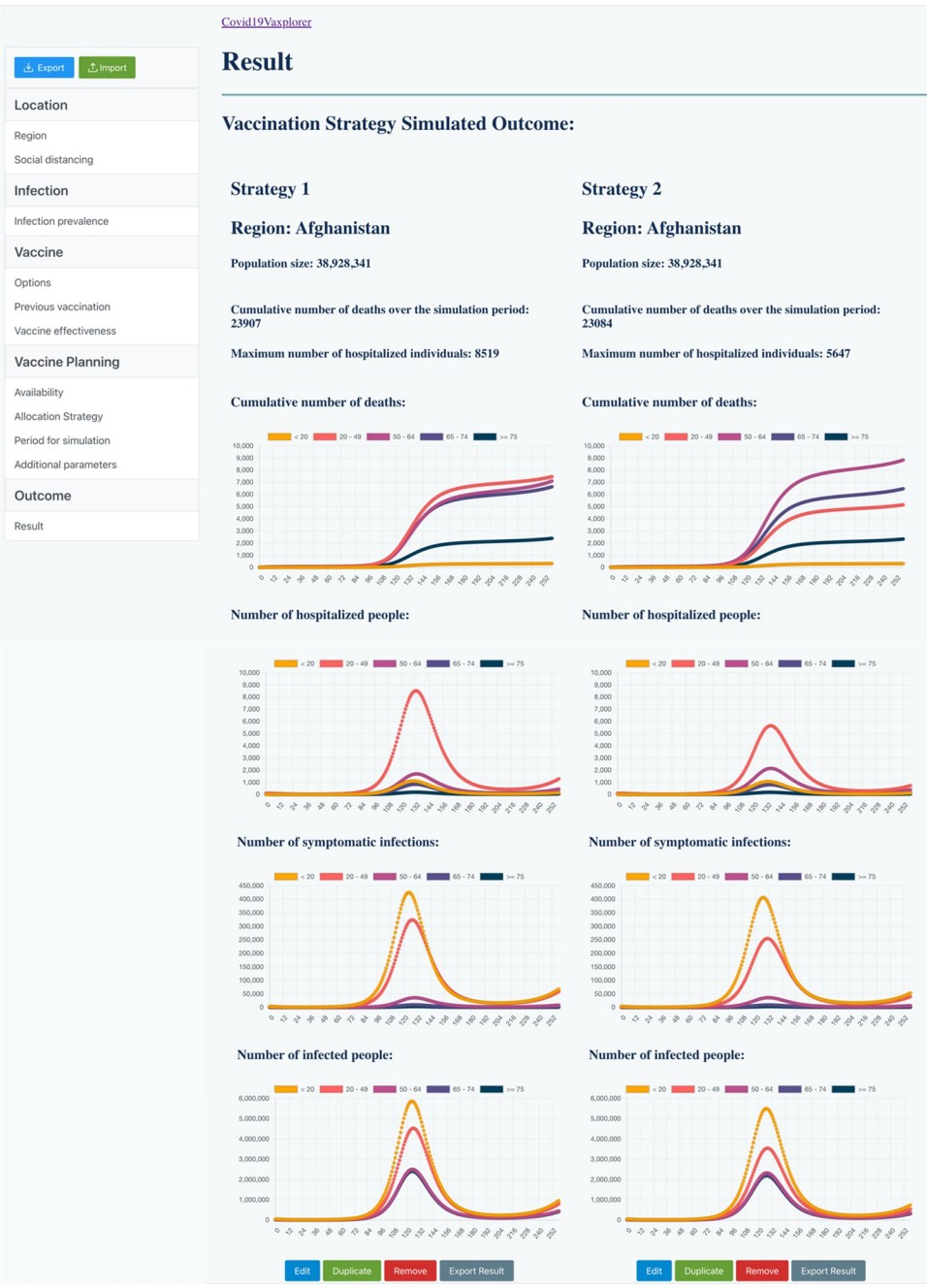

**Fig 8. Covid19Vaxplorer results page for comparing two strategies in a hypothetical situation in Afghanistan.**
Strategy 1, "boosters first" strategy, would result in 23907 cumulative deaths at the end of the simulation period and 8519 hospitalizations at the peak of the epidemic wave. In contrast, Strategy 2, "primary series first" would result in 23084 cumulative deaths and 5647 hospitalizations at the peak.

## Discussion

The COVID-19 pandemic has highlighted the need to use quantitative tools to take evidence-based decisions for vaccine distribution. As the pandemic has evolved into an endemic period, important questions remain regarding COVID-19 vaccine allocation. In particular, decision-

makers in LMIC, where large proportions of the population remain unvaccinated and or unboosted, might face difficult decisions regarding how to allocate COVID-19 vaccine doses: to give boosters to their high-risk populations or to vaccinate with a primary series their unvaccinated populations. Yet, many countries lack the resources to quantitatively compare different vaccination strategies. **Covid19Vaxplorer**, an online tool to simulate and compare different COVID-19 vaccination strategies, was developed to fill this gap, so that decision-makers around the world have access to a simple tool allowing them to make informed, evidence-based decisions.

We developed **Covid19Vaxplorer** with three main factors in mind. First, **Covid19Vaxplorer** needed to be free, so that anyone in the world could use it. Second, **Covid19Vaxplorer** needed to be online, so that it would be accessible for anyone with internet access. Finally, **Covid19Vaxplorer** needed to be easy to use, so that decision makers, with or without mathematical experience, could use it to compare potential vaccination strategies. This last point proved to be the most difficult to achieve, as we needed to strike a balance between a simple enough tool to be user-friendly and a realistic enough mathematical model to provide useful vaccine allocation comparisons.

Vaccine uptake depends on a variety of factors, and **Covid19Vaxplorer** was constructed with logistical factors in mind, e.g. number of available vaccines, types of vaccine products, regional constraints. Yet, as the pandemic has evolved, these logistical factors might have become less important while other social, cultural or behavioral factors, as vaccine hesitancy or misinformation, might have become more dominating. While we recognize that this is a limitation of **Covid19Vaxplorer**, we believe that this tool can still be useful for vaccination planning and for comparing vaccination strategies, as those sociological factors are likely to remain unchanged when comparing two vaccination strategies based on logistical factors (for example, when comparing vaccination strategies differing on the age of the vaccinated groups).

**Covid19Vaxplorer** is subject to several limitations. It might be challenging to obtain the required information to fill the initial conditions and other parameters to run **Covid19Vaxplorer**. While we have made every possible effort to provide the users with as many default parameters as possible, certain key numbers (e.g. current and past infection prevalence, the value of $R_0$ for the circulating strain) are location-specific and might be difficult to ascertain. **Covid19Vaxplorer** allows users to compare a few strategies at a time, but it cannot compare all the possible strategies. Doing so would require to optimize over all possible vaccine allocations, and this is too demanding computationally to do online.

In addition, as any mathematical model, we have done some simplifying assumptions, including: First, the model does not distinguish between first and second doses and encompass both under a primary series label. While it incorporates product-specific information about the length of vaccine-induced immunity and its effectiveness, it assumes a single duration of immunity for waned vaccinated individuals following a SARS-CoV-2 infection. Furthermore, immunity might differ by variants, (variants in the vaccines, variants in previous infections and variants in current infections) but **Covid19Vaxplorer** does not have the capability to model all of them. We assumed that two infection events would confer similar protection as a primary series vaccination, but it is possible that a single infection might suffice to be equally protective. **Covid19Vaxplorer** requires many input parameters for which there is considerable uncertainty. To mitigate this issue, it is advised to run **Covid19Vaxplorer** with many parameter sets, considering plausible ranges of parameters as a way to obtain more robust conclusions when comparing possible scenarios. New information regarding how vaccine and natural infection interact and shape immunity to COVID-19 continues to emerge. It is possible that our model does not accurately capture these interactions. It has been well established that different comorbidities affect COVID-19 severity (obesity, diabetes, hearth disease are all risk

factors for COVID-19) but the model does not include these factors in the hospitalization rates. However, because the model is open source and available for download, users with specific information for their population can alter the hospitalization rates to include these comorbidities. **Covid19Vaxplorer** is not fitted to any epidemic in particular, rather, it provides generally accepted values for the most common parameters. While the user can modify the parameters for their own region, it is not suited to make projections and it should not be used or interpreted in such a way. Rather, the value of **Covid19Vaxplorer** relies in its ability to allow users to make back-to-back comparisons of vaccination strategies.

There is a vast body of work that was developed to compare COVID-19 vaccination strategies. Most of the previous research has been devoted to analyze within-country or within-region vaccine allocation [17–19, 21, 23, 24, 26, 31, 33, 74–84], some has explored the optimal use of COVID-19 vaccines globally to reduce between- [20, 22, 85, 86], or within-country inequities [84, 87, 88]. **Covid19Vaxplorer** adds to previous work with the hope of providing those countries or regions with limited modeling capabilities with a tool for vaccine allocation comparison.

## Supporting information

**S1 Text. Additional model and tool description and parameters.**
(PDF)

**S1 Fig. Example of the social distancing window in Covid19Vaxplorer.** The user can select for each location a multiplier (using a sliding bar) representing the reduction in the number of contacts in that particular location.
(PNG)

**S2 Fig. Example of the previous vaccination windows in Covid19Vaxplorer.** There are five such windows, one per age group (group 3 is visualized here). In each window, the user inputs the proportion of that age group that has been previously vaccinated either with a primary series or with a booster for each vaccine product.
(PNG)

**S3 Fig. Example of the vaccine availability window in Covid19Vaxplorer.** The user inputs for each vaccine product the amount of product available and the vaccination rate (i.e. the number of doses to be distributed per day).
(PNG)

**S4 Fig. Example of the period for simulation window in Covid19Vaxplorer.** The tool provides a default end of simulation date based on the number of vaccines to be allocated and the vaccination rates (full details in Methods), but the user can change the default values.
(PNG)

**S5 Fig. Example of the additional parameters window in Covid19Vaxplorer.** Additional parameters for the model are displayed in this window. While Covid19Vaxplorer has default parameters, the user can modify all of them.
(PNG)

**S6 Fig. Tables with the distributions across age groups of Jansen, Convidecia and Vero Cell vaccines to be given as a primary series or as boosters under the strategies of "Boosters first" (Strategy 1) and "Primary series first" (Strategy 2) for the hypothetical example in Haiti.**
(PNG)

**S7 Fig. Covid19Vaxplorer results page for comparing two strategies in a hypothetical situation in Haiti assuming 80% of the population has pre-existing immunity.** Strategy 1, "boosters first" strategy, would result in 8,836 cumulative deaths at the end of the simulation period. In contrast, Strategy 2, "primary series first" would result in 6791 cumulative deaths. (PNG)

**S8 Fig. Tables with the distributions across age groups of the Jansen vaccine to be given as a primary series or as boosters under the strategies of "Boosters first" (Strategy 1) and "Primary series first" (Strategy 2) in a hypothetical scenario in Afghanistan.** (PDF)

**S1 Table. Table of default parameters used in the model.** (PDF)

**S2 Table. Table of default vaccine effectiveness values provided by Covid19Vaxplorer.** (PDF)

**S3 Table. Assumed distribution of COVID-19 prior vaccination in Afghanistan per age group and vaccine product.** (PDF)

## Acknowledgments

We thank Dr. Dobromir Dimitrov for helpful comments while developing this project.

## Author Contributions

**Conceptualization:** Laura Matrajt.

**Data curation:** Imelda Trejo.

**Formal analysis:** Imelda Trejo, Pei-Yao Hung.

**Funding acquisition:** Laura Matrajt.

**Investigation:** Imelda Trejo, Pei-Yao Hung, Laura Matrajt.

**Methodology:** Laura Matrajt.

**Supervision:** Laura Matrajt.

**Visualization:** Pei-Yao Hung, Laura Matrajt.

**Writing – original draft:** Laura Matrajt.

**Writing – review & editing:** Imelda Trejo, Pei-Yao Hung, Laura Matrajt.

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
