## [Decision Letter · Decision Letter 0]

20 Jul 2023

PGPH-D-23-01112

Covid19Vaxplorer: a free, online, user-friendly COVID-19 Vaccine Allocation Comparison Tool

Dear Dr. Matrajt

Thank you for submitting your manuscript to PLOS Global Public Health. After careful consideration, we feel that it has merit but does not fully meet PLOS Global Public Health’s publication criteria as it currently stands. Therefore, we invite you to submit a revised version of the manuscript that addresses the points raised during the review process.

We look forward to receiving your revised manuscript.

Kind regards,

Sizulu Moyo, MBCBH, MPH, PhD

Academic Editor

Journal Requirements:

1. Please note that PLOS GPH has specific guidelines on code sharing for submissions in which author-generated code underpins the findings in the manuscript. In these cases, all author-generated code must be made available without restrictions upon publication of the work. Please review our guidelines at https://journals.plos.org/plosone/s/materials-and-software-sharing#loc-sharing-code and ensure that your code is shared in a way that follows best practice and facilitates reproducibility and reuse.

2. Please note that your Data Availability Statement is currently missing the repository name and a direct link to access each database as the link has no uploaded files. 

If your manuscript is accepted for publication, you will be asked to provide these details on a very short timeline. We therefore suggest that you provide this information now, though we will not hold up the peer review process if you are unable.

3. Please declare all competing interests beginning with the statement "I have read the journal's policy and the authors of this manuscript have the following competing interests:"

4. Please provide separate figure files in .tif or .eps format and remove the embedded figures from the manuscrpt file.

5. We notice that your supplementary [figures/tables] are included in the manuscript file. Please remove them and upload them with the file type 'Supporting Information'. Please ensure that each Supporting Information file has a legend listed in the manuscript after the references list.

6. Figures 2-5 and S3 contains screenshots. We are not permitted to publish these under our CC-BY 4.0 license; websites are usually intellectual property and are copyrighted.This includes peripheral graphics of the web browsers. We ask that you please remove or replace it.

Additional Editor Comments (if provided):

Please review references and to ensure reasons for the reference.

Ensure that the underlying data are provided

Reviewers' comments:

Reviewer's Responses to Questions

**Comments to the Author**

1. Does this manuscript meet PLOS Global Public Health’s publication criteria? Is the manuscript technically sound, and do the data support the conclusions? The manuscript must describe methodologically and ethically rigorous research with conclusions that are appropriately drawn based on the data presented.

Reviewer #1: Partly

Reviewer #2: Yes

2. Has the statistical analysis been performed appropriately and rigorously?

Reviewer #1: N/A

Reviewer #2: Yes

3. Have the authors made all data underlying the findings in their manuscript fully available (please refer to the Data Availability Statement at the start of the manuscript PDF file)?

Reviewer #1: Yes

Reviewer #2: No

4. Is the manuscript presented in an intelligible fashion and written in standard English?

Reviewer #1: Yes

Reviewer #2: Yes

5. Review Comments to the Author

Reviewer #1: The article deals with a very important topic and the goal is worthy. The article should be improved in different ways, especially in the presentation of details. The model and the developed software have a lot of details that need to be explained. The authors probably devoted a lot of time and now they need to give more value and clarity to their work.

The authors must emphasize the aim of the article. For instance, we developed Covid19Vaxplorer in order to …?

Not clear why the authors mentioned the 183 regions in the abstract. I think there are more important aspects that can be mentioned/clarified in the abstract.

The results section in the abstract starts with “we usage two examples”. I think this section can be improved a lot, mentioning the model, software and the applications of this work.

The introduction must be improved. There are some related works regarding vaccination strategies and mathematical models. The authors should provide a good review of these in order to present a better background to the topic. The current version looks short on this matter.

The presentation of the model should be clearer. It would be very difficult for many users to understand the model. I suggest adding a Table that describes the different groups in detail and how they differentiate. Figure 1 is useful, but it seems not enough.

Page 4. Explain better "We model vaccination campaigns with a primary series or boosters. We do not model individual doses for primary series."

Page 4. Please explain the rationale to distribute the vaccines proportionally between groups. Make sense that recovered groups have the same rate?

Page 4. Please explain “Vaccinated individuals from compartments E_, A_, and P_ do no acquire any type of vaccine protection.”

Page 4. Typo Vij,, Vij. ??

Page 6. The authors mentioned: “Third, users are invited to specify planning-related parameters, including vaccine availability (what and when), and how vaccines will be allocated to different age groups within the population.”

Here I am confused since previously the authors mentioned that allocation of vaccines is proportional to the size of the populations. This and other details need much better explanation.

Page 6. The authors stated “Finally, users are presented with projections of key epidemic outcomes based on the information they input.” Please add details about what are these outcomes. It is quite difficult to grasp the aims of the developed software.

Page 7. The authors should explain how the input value of the Ro is used in the model. This is not clear. For instance, the authors mentioned that the duration of the infectious period is chosen by the user. What expression Ro is used? Comment on the fact that Ro is related to one infective person introduced in a fully susceptible population. How does the software handle the Ro as an input? This needs more explanation in the manuscript.

Page 7.

The authors wrote:

“Social distancing: User selects how people are practicing social distancing at each

location (home, work, school, community) if people are practicing social distancing,

ranging from “fully social distanced” to “not at all” social distanced. These inputs

are translated to the model as multipliers to each of these contact matrices”

This needs more details. For instance, fully social distanced translate on zero transmission rates?. Is not clear how the location (home,work,school,community) is integrated in the model presented in Figure 1.

Page 7. Authors wrote:

“Infection Prevalence: In this window, the user inputs the proportion of people in

each age group who have been previously infected (and who then will have some

protection against reinfection, symptoms and/or severe disease) and those who are

currently infected. The window provides default percentages for cumulative prior

infection as of November 2021 taken from [19].”

In some way these are the initial conditions for the 540 subpopulations. This is a very complex task that need further explanation. Some details about the default values should be added. For instance, as the authors mentioned each country has different percentages of vaccinated people. How the default option takes this into account.

Page 8. The authors wrote:

“Period for simulation: The user provides the beginning and end dates for the simulation.

Default end dates are provided based on the amounts of vaccine available and vaccination rates,”

Please explain more details about this process. Assuming a particular number of vaccines and a rate how the end date is computed. Due to the structure of the model (ODEs?) and the way vaccines are assigned this point is not clear.

Page 8. “4. Outcome: This is where the results are displayed and visualized”

Does the software include the epidemic curves by age groups and for any group?

Please add comments on the manuscript.

End of page 8. The authors wrote:

“For both strategies, vaccination starts with older age groups and goes in decreasing order by age. For example, under in Strategy 1, we would start vaccinating the oldest age group (group 5, 75 years old and older),”

Does always the vaccination strategy start with the oldest age group or is this just for the example on the Results section. Better explanation with more details needed.

Page 11. The authors wrote:

“then adults in group 3 (50-65 years old) and so on as vaccine availability permits.”

Please add detailed explanation. Is not clear if the vaccines can be allocated to different groups and with different proportions at the same time.

Page 11. The authors wrote:

“sensitivity analysis with 80% pre-existing immunity showed similar qualitative results with a strategy allocating primary series first outperforming the one prioritizing boosters”

Please add more details about all the parameter used in this scenario and add discussion. For instance, outperforming based on what? Number of deaths? number of cases? etc.

Page 11. Authors wrote:

“We consider a hypothetical situation in which Haiti would experience a new SARS-CoV-2 wave

with a variant with similar infectivity as the delta variant (R0 = 3) with 0.1% of the population

currently infected. ” Please explain how this is achieved in the software. It is not clear. The authors mean just assuming that R0=3 from the beginning of the simulation on the authors mean that the new variant is introduced at some point?

Page 11. The authors wrote:

“We assume that for all three vaccines, 50,000 individuals can be vaccinated daily.”

Please explain how the vaccines are distributed. How the model is modified such 50k vaccines can be allocated.

Page 15. The authors wrote:

“While it incorporates product-specific information about the length of vaccine-induced immunity and its effectiveness, it assumes a single duration of immunity for waned vaccinated individuals”

Not clear why the authors mention that the model assumes a single duration of immunity. Please revise this statement. The waning is modeled by a differential equation with a constant coefficient.

Page 15. Typo : “used or construed”

In summary, I think the article is valuable and deals with an extremely important topic. The authors need to explain much more details. The presentation of the model should be clearer. It would be very difficult for many users to understand the model. Probably the article needs to be extended to make a better presentation. Not sure if the journal has page limitations. There are many more limitations that the authors need to mention. For instance, the values of the parameters play an important role on the dynamics and they have uncertainty. This is just one example of limitations that can be added. Variants?

Reviewer #2: Review: Covid19Vaxplorer: a free, online, user-friendly COVID-19 Vaccine Allocation Comparison Tool

The manuscript describes a developed online tool proposed for use to assess the impact on different COVID-19 vaccination strategies on covid related outcomes that includes hospitalization and mortality. The authors use deterministic mathematical models.

Comments

1. This tool would probably have been more useful and relevant in the immediate post availability of COVID-19 vaccines and when the COVID-19 pandemic was still growing with high levels of deaths and hospitalizations. For various reasons vaccination uptake anywhere in the World has dramatically come down, and country priorities on COVID-19 vaccination have probably shifted.

2. Vaccination uptake depends on a number of factors including system factors, which seems what this paper is mostly focusing on, i.e. helping decision makers on vaccination strategies. During the early stages of the COVID-19 epidemic and when vaccines were first introduced, the prohibitive costs of vaccines, vaccine hording where some of the barriers to COVID-19 vaccination roll out in middle- and low-income countries. With COVID-19 vaccines being more readily available and cheaper across the world, the important factors that are most informative of the success of the outcomes of COVID-19 vaccination programmes are likely to be social, cultural, and behavioral factors, and not necessary the systems factors as mostly modelled in this paper.

3. Although the authors indicated that these models should not be used for predictive purposes, but for comparing different scenarios, it would have been useful to fit the models to real-life observed scenarios, i.e. test the models on real empirical data to test the reasonableness of the assumptions being made, and how well these models fit to observed experiences.

4. It is not clear from the model if known epidemiological concepts were incorporated in outcomes such as the risk of death and/or serious illness. For example, age was shown to be a risk factor for COVID-19 outcomes, with higher COVID-19 associated mortality and hospitalization being associated with older age. Comorbidities such as hypertension, diabetes, and those with compromised immune systems were also shown to be associated with COVID-19 outcomes.

5. There are minor comments highlighted on the manuscript.

6. PLOS authors have the option to publish the peer review history of their article (what does this mean?). If published, this will include your full peer review and any attached files.

**Do you want your identity to be public for this peer review?** For information about this choice, including consent withdrawal, please see our Privacy Policy.

Reviewer #1: No

Reviewer #2: No

---

## [Decision Letter · Decision Letter 1]

13 Nov 2023

PGPH-D-23-01112R1

Covid19Vaxplorer: a free, online, user-friendly COVID-19 Vaccine Allocation Comparison Tool

Dear Dr. Laura Matrajt

Thank you for submitting your  revised manuscript to PLOS Global Public Health. After careful consideration, we feel that it has merit but does not fully meet PLOS Global Public Health’s publication criteria as it currently stands. Therefore, we invite you to submit a revised version of the manuscript that addresses the points raised during the review process.

We look forward to receiving your revised manuscript.

Kind regards,

Sizulu Moyo, MBCBH, MPH, PhD

Academic Editor

Journal Requirements:

Additional Editor Comments (if provided):

Reviewers' comments:

Reviewer's Responses to Questions

**Comments to the Author**

1. If the authors have adequately addressed your comments raised in a previous round of review and you feel that this manuscript is now acceptable for publication, you may indicate that here to bypass the “Comments to the Author” section, enter your conflict of interest statement in the “Confidential to Editor” section, and submit your "Accept" recommendation.

Reviewer #1: (No Response)

2. Does this manuscript meet PLOS Global Public Health’s publication criteria? Is the manuscript technically sound, and do the data support the conclusions? The manuscript must describe methodologically and ethically rigorous research with conclusions that are appropriately drawn based on the data presented.

Reviewer #1: Yes

3. Has the statistical analysis been performed appropriately and rigorously?

Reviewer #1: N/A

4. Have the authors made all data underlying the findings in their manuscript fully available (please refer to the Data Availability Statement at the start of the manuscript PDF file)?

Reviewer #1: Yes

5. Is the manuscript presented in an intelligible fashion and written in standard English?

Reviewer #1: Yes

6. Review Comments to the Author

Reviewer #1: The authors improved the article and now is clearer. Due to the large number of details, I understand that it is difficult to cover all or easy to miss aspects that readers would be interested.

The authors added in the new version very general references related to mathematical models that have been used for the COVID pandemic. However, since one specific aim of this work is to compare vaccine allocation strategies the references and discussion should be related to this specific aim.

In the modeling contacts section, it says that the user must select the multiplier (between 0 and 1) related to the social contact. One represents full contact. Please add further detailed explanation for this. Many readers can be confused. There are 4 social contact matrices and a force of infection that depends on the values of these matrices.

In the modeling initial conditions please mention how many values the user needs to provide. Please remark or comment all the difficulties that this aspect might have for a user. This point is crucial for the outcomes of the simulation.

1- Please mention the assumed proportion of asymptomatic in this section or clarify that is also an input provided by the user.

Page 7. "If the vaccine is used as a booster, the user also provides the age and

previously vaccinated groups where the booster will be applied." This phrase is not clear.

Please explain how the allocation of strategies per day is incorporated in the model. What type of ODE model it is. These small details can help readers to grasp the methodology and the suitability for each region. Looking and explaining Eq. (4) can help.

Page 8. “However, we assume that two infection events act as a primary

series vaccination. “Please add a brief explanation and/or one reference for this assumption.

In the discussion section I think is fair to add further limitations. For instance, the user needs a good knowledge of the parameters of the model. In addition, the user needs to have knowledge of the initial conditions (please add how many values are needed as an input). Also, the authors should emphasize or remark that the number of strategies is quite large and for a user might be impossible to check all the potential strategies. The examples provide few strategies. I understand that including more examples could be demanding, but mentioning this can help the readers.

The aim of the article is worthy and important. The aims are very challenging due to all the factors that affect the outcomes of the simulations.

In the supplementary part please add the Ro as explicit as possible. This helps the readers.

Page 36. Typo experience.

7. PLOS authors have the option to publish the peer review history of their article (what does this mean?). If published, this will include your full peer review and any attached files.

**Do you want your identity to be public for this peer review?** For information about this choice, including consent withdrawal, please see our Privacy Policy.

Reviewer #1: No

---

## [Editor Report · Decision Letter 2]

13 Dec 2023

Covid19Vaxplorer: a free, online, user-friendly COVID-19 Vaccine Allocation Comparison Tool

PGPH-D-23-01112R2

Dear Dr Matrajt

Thank you for addressing the comments raised. We are pleased to inform you that your manuscript 'Covid19Vaxplorer: a free, online, user-friendly COVID-19 Vaccine Allocation Comparison Tool' has been provisionally accepted for publication in PLOS Global Public Health.

Best regards,

Sizulu Moyo, MBCBH, MPH, PhD

Academic Editor